# What Is the Significance of Lysosomal-Mediated Resistance to Imatinib?

**DOI:** 10.3390/cells12050709

**Published:** 2023-02-23

**Authors:** Petr Mlejnek

**Affiliations:** Department of Anatomy, Faculty of Medicine and Dentistry, Palacky University Olomouc, Hnevotinska 3, 77515 Olomouc, Czech Republic; mlejnek_petr@volny.cz

**Keywords:** STI571, hydrophobic weak-base drugs, lysosomal drug sequestration, drug resistance mechanisms, tyrosine kinase inhibitors, cancer

## Abstract

The lysosomal sequestration of hydrophobic weak-base anticancer drugs is one proposed mechanism for the reduced availability of these drugs at target sites, resulting in a marked decrease in cytotoxicity and consequent resistance. While this subject is receiving increasing emphasis, it is so far only in laboratory experiments. Imatinib is a targeted anticancer drug used to treat chronic myeloid leukaemia (CML), gastrointestinal stromal tumours (GISTs), and a number of other malignancies. Its physicochemical properties make it a typical hydrophobic weak-base drug that accumulates in the lysosomes of tumour cells. Further laboratory studies suggest that this might significantly reduce its antitumor efficacy. However, a detailed analysis of published laboratory studies shows that lysosomal accumulation cannot be considered a clearly proven mechanism of resistance to imatinib. Second, more than 20 years of clinical experience with imatinib has revealed a number of resistance mechanisms, none of which is related to its accumulation in lysosomes. This review focuses on the analysis of salient evidence and raises a fundamental question about the significance of lysosomal sequestration of weak-base drugs in general as a possible resistance mechanism both in clinical and laboratory settings.

## 1. Introduction

Chemotherapy remains the standard form of treating malignant tumours, and its role has grown significantly with the introduction of targeted chemotherapy, which minimises the negative effects on healthy cells [1]. Unfortunately, the efficacy of even the best available targeted chemotherapy is beleaguered by the emergence of drug resistance, the intrinsic resistance that exists at the outset of treatment, and acquired resistance, which develops during treatment. Intrinsic drug resistance is either associated with altered absorption, distribution, metabolism, and excretion resulting in suboptimal serum drug levels, or with pre-existing mutations in molecular targets and the genes that control proliferation and tumour cell death. Acquired resistance is caused by a number of mechanisms that may include (a) mutation in drug targets; (b) the activation of the genes that are involved in drug metabolism and detoxification; (c) alteration in cell death pathways; (d) the deregulation of cell cycle checkpoints; (e) enhanced drug-induced DNA repair; (f) decreased intracellular drug accumulation; (g) the effect of the tumour microenvironment; (h) the epithelial–mesenchymal transition; and (i) the activation of pro-survival signalling [2,3,4,5,6,7]. 

The lysosomal sequestration of drugs is increasingly mentioned as another key mechanism of resistance [8,9,10,11,12,13], sometimes considered an intrinsic resistance mechanism and sometimes acquired. 

## 2. Lysosomal-Mediated Drug Resistance 

Lysosomes, which have multiple cellular functions, are also recognised as mediators of drug resistance [8,9,10,11,12,13,14,15,16,17]. Given their acidic pH (4.5–6) relative to the cytosolic pH, which is close to neutral (7.2), they trap large amounts of hydrophobic weak-base drugs passively [18,19]. It is assumed that this phenomenon, also referred to as lysosomal drug sequestration, simultaneously reduces the drug availability at the target sites and thus decreases its cytotoxic effects [8,9,10,11,12,13,20,21,22,23,24]. A brief scheme of the mechanism of drug resistance mediated via passive lysosomal sequestration is given in Figure 1.

**Thought** **experiment:** The passive lysosomal sequestration of IM was investigated, and a quantitative analysis of its distribution was carried out in vitro. K562 cells (5 × 10^5^ cells/mL; cell average diameter = 18 μm) were placed in a 10 mL growth medium containing 1 μM IM. The molar amount of IM in the system can be described as follows:n_T_ = n_M_ + n_C_ + n_L_ (I)where n_T_ is the total molar amount of IM added to the growth medium; n_M_ is the molar amount of IM that remains in the medium after equilibrium has been established; n_C_ is the molar amount of IM in the cytosol after equilibrium has been established without lysosomal sequestration; and n_L_ is the molar amount of IM in lysosomes after equilibrium has been established. The molar amount of IM in “compartment i” can be expressed as the sum of the concentration of uncharged and charged forms using Equation (II).n_i_ = V_i_ (IM_i_ + IMH^+^_i_) (II)where n_i_ is the molar amount of IM; IM_i_ is the concentration of the uncharged form; IMH ^+^ _I_ is the concentration of the charged form; and V_i_ is the volume of the appropriate compartment. Indexes M, C, and L stand for medium, cytosol, and lysosome, respectively. A combination of I and II gives: n_T_ = V_M_([IMH]^+^ _M_ + [IM]_M_) + V_C_([IMH]^+^ _C_ + [IM]_C_) + V_L_([IMH]^+^ _L_ + [IM]_L_) (III)Given that an uncharged form of IM can freely diffuse across cell membranes, the following applies:[IM]_M_ = [IM]_C_ = [IM]_L_ (IV)A combination of III and IV gives: n_T_ = V_M_([IMH]^+^ _M_ + [IM]_L_) + V_C_([IMH]^+^ _C_ + [IM]_L_) + V_L_([IMH]^+^ _L_ + [IM]_L_) (V)n_T_ = (V_M_ + V_C_ + V_L_)[IM]_L_ + V_M_[IMH]^+^ _M_ + V_C_[IMH]^+^ _C_ + V_L_[IMH]^+^ _L_ (V)where V_M_ = 10mL, V_C_ = 15.3 μL (total cellular volume of 5 × 10^6^ K562 cells), and V_L_ = 0.03–0.05V_C_ = 0.46–0.77 μL (total volume of lysosomes assuming that they represent 3–5% of total cellular volume [25]).Given that V_M_ >>V_C_ and V_M_ >>V_L_ = > V_M_ + V_C_ + V_L_~V_M_, thenn_T_ = V_M_[IM]_L_ + V_M_[IMH]^+^ _M_ + V_C_[IMH]^+^ _C_ + V_L_[IMH]^+^ _L_ (VI) The relationship between the charged and uncharged forms of IM is given by equation VII:K_a_ = ([IM] [H^+^])/[ IMH^+^] (VII) which can be rewritten as follows:pK_a_ = log([IMH]^+^ /([IM][H]^+^)) (VII) which can be rewritten as follows:[IMH]^+^ = [IM]10exp(pK_a_ − pH) (VII)A combination of VI and VII gives: n_T_ = V_M_[IM]_L_ + V_M_[IM]_M_10exp(pK_a_ − pH_M_) + V_C_[IM]_C_10exp(pK_a_ − pH_C_) + V_L_[IM]_L_10exp(pK_a_ − pH_L_) (VIII)Given that we assume pH_M_ = pH_C_, thenn_T_ = V_M_[IM]_L_ + V_M_[IM]_L_10exp(pK_a_ − pH_M_) + V_C_[IM]_L_10exp(pK_a_ − pH_M_) + V_L_[IM]_L_10exp(pK_a_ − pH_L_) (IX)n_T_ = [ IM]_L_(V_M_ + V_M_10exp(pK_a_ − pH_M_) + V_C_10exp(pK_a_ − pH_M_) + V_L_10exp(pK_a_ − pH_L_))[IM]_L_ = n_T_/(V_M_ + V_M_10exp(pK_a −_ pH_M_) + V_C_10exp(pK_a_ − pH_M_) + V_L_10exp(pK_a_ − pH_L_))Given that IM pK_a_ = 7.84, pH_C_ = pH_M_ = 7.3, and pH_L_ = 4.5, then[IM]_L_ = n_T_/(4.5V_M_ + 3.5V_C_ + 2188V_L_) = > [IM]_L_ = 0.22 μM (V_L_ = 0.03V_C_); [IM]_L_ = 0.21 μM (V_L_ = 0.05V_C_)[IMH]^+^_M_ = 3.5[IM]_L_ = > [IMH]^+^_M_ = 0.77 μM (V_L_ = 0.03V_C_); [IMH]^+^_M_= 0.74 μM (V_L_ = 0.05V_C_)[IMH]^+^_C_ = 3.5[IM]_L_ = > [IMH]^+^_C_ = 0.77 μM (V_L_ = 0.03V_C_); [IMH]^+^_C_ = 0.74 μM (V_L_ = 0.05V_C_)[IMH]^+^_L_ = 2188[IM]_L_ = > [IMH]^+^_L_ = 481.4 μM (V_L_ = 0.03V_C_); [IMH]^+^_C_ = 459.5 μM (V_L_ = 0.05V_C_)n_L_ = V_L_([IMH]^+^_L_+[IM]_L_) = > n_L_ = 0.221 nM (V_L_ = 0.03V_C_); n_L_ = 0.382 nM (V_L_ = 0.05V_C_)n_L_/n_T_ = 0.022 (V_L_ = 0.03V_C_); n_L_/n_T_ = 0.038 (V_L_ = 0.05V_C_)These results show that lysosomes can accumulate 2.2–3.8% of the total IM amount. □

The passive lysosomal sequestration of hydrophobic weak-base drugs can be enhanced through two mechanisms that may further increase drug resistance. First, by enlarging the lysosomal compartment through fusion, which can be heterotypic (lysosome–endosome), homotypic (lysosome–lysosome), or through biogenesis [26,27,28,29,30], and second, via the active transport of the drug into the lysosomal lumen mediated by ABC transporters or the copper transporter [31,32,33,34]. Further, the combination of passive lysosomal sequestration with lysosomal exocytosis is also thought to decrease anticancer drug efficiency [35,36]. An overview of putative lysosomal-mediated mechanisms of drug resistance is depicted in Figure 2. 

However, no lysosome-mediated resistance mechanism has been clearly demonstrated, even in the case of anthracyclines, the best-studied of the weak-base antitumor drugs in vitro. Most studies confirm their sequestration, which affects their concentration at the target site. Others show the opposite [20,21,22,23,24,37,38,39,40]. It should be noted here that confirmatory studies are mostly based on image analyses of sensitive and resistant cells [20,21,22,23,24,37]. However, the fluorescence intensity of anthracyclines is not a simple linear function of concentration, making such analyses questionable [39,40,41]. Overall, despite inconsistencies in in vitro experiments and the incomparability of in vivo and in vitro conditions [42,43,44], a number of researchers suggest that lysosomal sequestration could reduce the effectiveness of weak-base antitumor drugs in clinical practice [9,10,11,12,13,24,33,34]. If so, this would be a serious problem, as many conventional and targeted chemotherapies fall into the category of weak-base drugs (Table 1). Fortunately, a critical evaluation of the available data shows that this is far from the case. We will demonstrate this here using the example of imatinib mesylate (IM, Glivec, Novartis, Basel, Switzerland), the first rationally developed inhibitor targeting Bcr-Abl tyrosine kinase (TK) [45]. IM is well suited for this purpose for a number of reasons. First, it is a typical hydrophobic weak-base drug (Table 1) that extensively accumulates in the lysosomes of cancer cells [44,46,47,48]. Second, there is a large volume of data describing the various mechanisms of resistance to IM in in vitro experiments [34,49,50,51]. Thirdly, there is extensive clinical experience with it, both in the treatment of CML and gastrointestinal stromal tumours (GISTs), including well-described clinically relevant resistance mechanisms [52,53,54,55,56,57,58,59,60,61]. 

## 3. Resistance to Imatinib in Laboratory Experiments

### 3.1. Lysosomal Sequestration of Imatinib and Drug Resistance In Vitro

At the beginning of this review, with a thought experiment, we sought to demonstrate how much IM accumulates in the lysosomes of tumour cells and how this accumulation affects its concentration in the cytosol (i.e., target site) and thus its antiproliferative and proapoptotic effects (Figure 1). The experiment was carried out with Bcr-Abl + K562 cells with an average diameter of 18 μm, at a density of 5 × 10^5^ cells/mL, in a growth medium with a total volume of 10 mL containing 1 μM IM (pKa = 7.84) at pH = 7.3. For simplicity, we assumed that only the uncharged form of IM can freely pass through the membranes; the cells were assumed to have an ideal spherical shape, with intracellular pH = 7.3 and lysosomal pH = 4.5. In addition, according to the literature, we assumed that the lysosomal volume represents 3–5% of the total cellular volume [25]. After implementing these assumptions, which, although simplified, are very close to real experimental conditions, the following findings were observed (Figure 1): (A) The lysosomal sequestration of IM, which reduces drug concentration at the target site (=cytosol), simultaneously decreased the extracellular drug concentration. (B) In lysosomes, the concentration of the charged form of IM ([IMH^+^]) was more than 2000 times greater than that of the uncharged form ([IM]). (C) In lysosomes, the total concentration of IM ([IMH^+^] + [IM]) was almost 500 times greater than that of the cytosol and medium. (D) Acidic pH in lysosomes increased the solubility of IM and this facilitated its extensive accumulation in this compartment. (E) Despite the latter, only approximately 2.2–3.8% of the total IM accumulated in this compartment (assuming lysosomes represent 3–5% of the total cell volume). (F) Thus, the passive lysosomal sequestration of IM could reduce its concentration at the target site by approximately 2.2–3.8% (assuming lysosomes represent 3–5% of the cellular volume).

These results hardly support the idea that the passive lysosomal sequestration of IM, even if extensive, can mediate resistance to this drug. This conclusion was also drawn in our experimental results [29,44]. Further, the results derived from experimental data showed that approximately 2% of the total amount of IM accumulated in the lysosomes of K562 cells under conditions very close to those of the thought experiment [44]. 

### 3.2. Selection of Bcr-Abl-Positive Cell Lines with Decreased Sensitivity to Imatinib

Almost immediately with the introduction of IM for the treatment of CML, resistant cells were selected and characterised in in vitro experiments. Mahon et al. found different mechanisms leading to IM resistance [49]. These included an increased expression of Bcr-Abl tyrosine kinase associated with the amplification of the *Bcr-Abl* gene in Baf/BCR-ABL-r, LAMA84-r, and AR230-r cell lines. In the LAMA84-r cell line, they also observed an increased expression of the ABCB1 transporter. Finally, a molecular mechanism of resistance was found in the K562 cell line, which was neither related to the increased expression of the Bcr-Abl oncoprotein nor to the mutations in its kinase domain [49]. This resistance mechanism has been attributed to the acquisition of compensatory mutations in genes other than Bcr-Abl [49]. Weisberg and Griffin identified the increased expression of the Bcr-Abl oncogene as the cause of the resistance [50]. Another research group also found the increased expression of the *Bcr-Abl* gene to be a cause of resistance to IM [51]. In addition, the mechanisms of resistance to IM, collectively called Bcr-Abl-independent mechanisms, were identified. Specifically, resistance to IM was found to be due to the increased expression of Src kinases, which, among other things, affect the proliferation and survival of CML cells [62,63]. Although research on resistant cells selected with gradually increasing concentrations of IM has continued and yielded some new findings elucidating the changes associated with the development of resistance [64,65], to the best of our knowledge, no IM-resistant cells have been generated whose mechanism of resistance was lysosome-mediated.

### 3.3. ABCA3-Enhanced Lysosomal Sequestration of Imatinib Decreases Sensitivity to This Drug

Chapuy et al. were the first to publish a detailed laboratory study describing lysosome-mediated resistance to IM, where the passive sequestration of IM was enhanced by the lysosomal ABCA3 transporter [34] (Figure 2). This study reported important findings that CD34-positive progenitors from CML samples, as well as a side population of Bcr-Abl-positive cells from untreated patients, strongly express intracellular ABCA3 [34]. These authors demonstrated that ABCA3 is localised to the lysosomal membrane and multivesicular bodies. In addition, they showed that ABCA3 expression level is critical for the sensitivity of CML cells to IM in vitro. Finally, they demonstrated that lysosomal accumulation capacity greatly increases with ABCA3 expression for lysotracker, a fluorescent lysosomal probe, and to a lesser extent for IM [34]. Based on these findings, they concluded that the facilitation of the lysosomal sequestration of IM by ABCA3 may contribute to resistance to this drug [34]. Although these conclusions are generally accepted, the authors provided no direct evidence that the lysosomal sequestration of IM enhanced via the ABCA3 transporter reduces its concentration at target sites [34]. 

Another very interesting discovery from the same research group was that tyrosine kinase inhibitors including IM stimulate ABCA3 expression via the embryonic stem cell-associated transcription factor SALL4 in both model CML cells and primary CML cells [66]. From these results, one would expect that the CML cell lines exposed to gradually increasing concentrations of IM would inevitably become resistant due to the greater expression of ABCA3 on the lysosomal membrane [34]. Although resistant cells from the CML cell lines have been generated in this way in many laboratories, to the best of our knowledge, the increased expression of ABCA3 has never been found as a possible cause of resistance to IM. Unfortunately, the answer to the question of why it was impossible to create CML cells with this resistance mechanism remains open.

ABCB1 (P-glycoprotein, MDR1) is another ABC transporter most commonly hypothesised to increase the passive lysosomal sequestration of some anticancer drugs (Figure 2). ABCB1 was originally described as the first membrane transporter mediating the acquired drug resistance in animals [67,68]. It is well documented that ABCB1 functions as a plasma membrane ATP-dependent efflux pump for a variety of structurally diverse anticancer drugs [69]. However, some researchers began to point to the possibility that ABCB1 could also be expressed in cellular organelles such as lysosomes and contribute to drug resistance through an altered intracellular drug distribution [31,70]. Although these considerations relate to DNR and not IM, it is apposite to discuss them here. The main reason is that these considerations are often mistaken for experimentally proven findings. One of the serious caveats to these considerations, namely the fact that the functional expression of ABCB1 on the lysosomal membrane has not yet been demonstrated, was recently formulated in the work of Szakacs and Abele [71]. However, to the best of our knowledge, no study has yet reported ABCB1 expressed on the lysosomal membrane as a cause of IM resistance.

### 3.4. Lysosomal Sequestration of Imatinib and Its Concentration at Target Sites

There are other studies that relate to lysosomal-mediated resistance to IM. These focus on the mechanism of drug resistance, i.e., a decrease in drug concentration at target sites due to sequestration by lysosomes. According to assumptions, we should measure a higher inhibitory effect on oncogenic signalling when the sequestration of the studied drug is blocked compared with unblocked sequestration (Figure 1c). 

Burger et al. were the first to investigate the lysosomal sequestration of IM and its relationship to oncogenic signalling [48]. Although around 70–80% of IM was found to accumulate in lysosomes, blocking its lysosomal sequestration had no effect on c-Kit signalling in IM-sensitive GIST-1 cells [48]. In our laboratory, we addressed this issue with chronic myeloid leukaemia K562 cells. We obtained similar results, namely that the inhibition of the lysosomal sequestration of IM had no effect on oncogenic Bcr-Abl signalling, even though 50–60% of intracellular IM was accumulated in the lysosomes of K562 cells [44]. Additionally, since these are extremely important results, one more fact is worth noting in relation to sunitinib (SUN) a multitargeted TKI, which is also a weak-base drug (Table 1). Gotink et al. worked with two tumour SUN-resistant cell lines, where lysosomal sequestration was identified as the mechanism of resistance [72]. Here, too, it was clearly shown that the lysosomal sequestration of SUN had no effect on the signalling of p-Akt and p-ERK ½ [72]. 

These results do not confirm but rather refute the hypothesis that the lysosomal sequestration of the drug actually reduces its availability at target sites. Our analysis is consistent with these experimental findings [44]. In this study, we pointed out that the lysosomal sequestration of an anticancer drug can significantly reduce its concentration at target sites only when it simultaneously reduces its extracellular concentration, which is theoretically possible but very unlikely given the in vitro conditions, even in the case of drugs with suitable physicochemical properties, such as IM (Table 1; Figure 1), [44]. Our results showed that passive lysosomal sequestration can be enhanced by enlarging the lysosomal compartment. Unlike other authors, however, we did not observe lysosomal biogenesis but lysosomal fusion, which also leads to an increase in the lysosomal compartment [29], (Figure 2). However, this effect is significant only for high micromolar IM concentrations, which are clinically irrelevant [29,44]. It should be mentioned that we failed to detect lysosomal biogenesis induced by submicromolar or low micromolar concentrations of hydrophobic weak-base drugs including IM in various cancer cell lines [29,44,73,74]. 

### 3.5. Lysosomal Alkalinisation and Cell Sensitivity to Imatinib

It is assumed that the passive lysosomal sequestration of hydrophobic weak-base drugs can be prevented using specific noncytotoxic alkalising agents, which will lead to the reversal of lysosome-mediated drug resistance [13,75] (Figure 1). The most effective, though quite toxic, alkalising agents include vesicular H^+^-ATPase (v-ATPase) inhibitors, bafilomycin A1 (BafA1) and concanamycin A (ConA) [76,77,78], and chloroquine (CQ) or its derivative hydroxychloroquine (HCQ), which have low cytotoxicity [13,75]. Indeed, CQ and HCQ are approved drugs by the Food and Drug Administration (FDA) to prevent and treat malaria [79]. Not surprisingly, CQ was found to enhance the cytotoxic effect of IM in model CML and GIST cell lines, including resistant ones [80,81,82,83,84]. Detailed studies have shown that, in addition to apoptosis, IM induces the autophagy-related survival mechanism in CML primary cells, CML blast crisis cell lines, and Bcr-Abl-expressing myeloid precursor cells [80,81,82]. The same applies to the GIST cell lines, where IM also induces autophagy [83,84]. The authors agreed that the inhibition of autophagy through CQ leads to the elimination of tumour cells that are resistant to IM [80,81,82,83,84]. These conclusions are consistent with those of other authors [85,86]. To the best of our knowledge, no study has yet reported the inhibition of IM sequestration due to the alkalinisation of the acidic pH of lysosomes using BafA1 (ConA) or CQ (HCQ) as the direct agent potentiating the cytotoxic effect of IM in cancer cells. 

### 3.6. Lysosomal Exocytosis and Cell Sensitivity to Imatinib

The combination of lysosomal sequestration with exocytosis is one contributory mechanism in weak-base anticancer drug resistance [35,75,87,88] (Figure 2). However, to date, no study appears to have reported lysosomal exocytosis (combined with lysosomal sequestration) as a mechanism in IM resistance in cancer cells.

## 4. What Conclusions Can Be Drawn from Laboratory Experiments?

Extensive laboratory experiments have shown that no IM-resistant cells have been found, the mechanism of resistance of which is clearly lysosomal-mediated [44,48,49,50,51]. Importantly, the lysosomal sequestration of IM has never been shown to lead to a decrease in its concentration at target sites in in vitro experiments [44,48]. On the contrary, the lysosomal sequestration of IM has no effect on its target site concentration [44,48]. Additionally, no study has yet identified the inhibition of IM sequestration due to the alkalinisation of the acidic pH of lysosomes as a direct cause of the cytotoxic potentiating effect of this drug in cancer cells [80,81,82,83,84].

One exception, might be a study where the lysosomal sequestration of IM was enhanced by the ABCA3 transporter with reduced susceptibility of CML cell lines to this drug [34]. Even here, however, no causal relationship between ABCA3-mediated IM sequestration into lysosomes and a decrease in drug concentration at target sites was demonstrated [34]. Moreover, no one has yet been able to establish the cells resistant to IM, whose mechanism of resistance is related to lysosomal sequestration mediated by the increased expression of the ABCA3 transporter [34,66].

## 5. Clinical Resistance to IM

### 5.1. Clinical Resistance to IM in Patients with CML 

Resistance to IM in CML is also assessed with respect to other criteria such as disease stage (chronic versus advanced phase); the type of response detectable at a hematologic, cytogenetic, or molecular level; and the time when it was achieved given by The National Comprehensive Cancer Network and the 2013 European LeukemiaNet guidelines [56]. For the purposes of the present study, we will limit ourselves to a simplified overview of primary and secondary resistance mechanisms.

Several intrinsic mechanisms that may contribute to IM resistance have been described. First, CYP3A4 should be noted, which belongs to the cytochrome p450 superfamily of metabolic proteins and is involved in IM clearance. Although CYP3A4 cannot be considered a typical mediator of resistance, its activity significantly affects IM bioavailability. CYP3A4 activity depends on both individual variability and the co-administration of certain drugs [54,56]. 

Another frequently mentioned factor is the organic cation transporter-1 (OCT-1), which functions as an influx pump for IM and thus may influence its intracellular concentration [54,56]. Although some laboratory results suggest that IM may not be transported by OCT-1 [48], in clinical practice, the expression level of this transporter in primary CML samples serves as an important prognostic indicator of response to IM treatment at diagnosis [89].

The binding of IM to α1-acid glycoprotein (AGP), which could reduce its availability, is a somewhat contradictory mechanism of intrinsic resistance. Although initial studies showed that AGP binds IM and therefore reduces the availability of free/active drugs [90], others have not confirmed that the binding of IM to AGP is a possible mechanism of resistance [91,92]. The existing mutations in the kinase domain of the Bcr-Abl oncoprotein may also contribute to primary resistance, although rarely.

The special properties of CML stem cells are a far more significant factor contributing to primary resistance to IM. Here, a variety of mechanisms were identified that render them resistant to IM and may include a decreased expression of OCT-1, an increased expression of drug efflux transporters ABCB1 and ABCG2, or the status of quiescence [54].

The most common causes of acquired resistance are mutations in the kinase domain of the Bcr-Abl1 oncoprotein. They comprise 12 residues (M244, G250, Q252, Y253, E255, V299, F311, T315, F317, M351, F359, and H396) and are responsible for 50–90% of relapse in CML patients. An increased expression of Bcr-Abl1, either due to the duplication of the entire Ph chromosome, the amplification of the Bcr-Abl1 gene, or the increased transcription of an oncogene, is another clinically described mechanism of resistance [53,54,56,57,93].

Other clinically important mechanisms of acquired resistance are collectively called Bcr-Abl-independent mechanisms. These are numerous and include the aberrant activation of signalling pathways or factors that control CML cell proliferation and survival, such as Raf/Mek/Erk, PI3K/Akt, STAT3, Src kinases, hypoxia-inducible factor 1α, Wnt-β-catenin, and many others [54,56,94,95].

### 5.2. Clinical Resistance to IM in Patients with GISTs

Approximately 20% of patients with GISTs do not initially respond to IM therapy. The most common causes are mutations in *KIT* and, less often, mutations in *PDGFR*. *KIT* amplification has been reported as another reason for initial treatment failure. Alterations in independent KIT or PDGFR pathways or increased drug metabolism are other important innate mechanisms of resistance to IM [55,58,59,60,61]. 

In acquired resistance, secondary mutations in *KIT* or *PDGFR* are predominant. These are followed by an increased expression of KIT due to gene amplification. The important mechanisms of acquired resistance also include the activation of alternative signalling pathways or mutations in *KIT* or *PDGFR* that do not affect IM binding [55,58,59,60,61]. Alternatively, it is speculated that the increased expression of ABC transporters such as ABCB1 could contribute to IM resistance [58,61].

### 5.3. ABCA3-Mediated Resistance to IM in Clinics 

The increased expression of ABCA3 appears to be associated with a resistant phenotype only in acute myeloid leukaemia (AML) patients but not in CML patients [96,97,98]. This may seem surprising, especially in connection with the studies of the research group of Professor G. Wulf (Georg August University, Goettingen) [33,34,66,98]. It is necessary to note that even in the case of ABCA3-mediated resistance in AML patients, there is no direct evidence that this resistance is related to a decrease in drug concentration at target sites [96,97,98]. These findings suggest that ABCA3 expression might have prognostic relevance rather than being directly linked to the mechanism of drug resistance [99]. To the best of our knowledge, ABCA3 has never been mentioned in association with clinical resistance to IM in GIST patients. At this point, it should be noted that even in other malignancies, high ABCA3 expression is understood as a negative prognostic factor rather than a mediator of drug resistance in clinics [100,101].

### 5.4. Clinical Application of CQ or HCQ in Combination with IM 

Both CQ and HCQ are used in clinical practice to increase the antitumor efficacy of chemotherapy, both untargeted and targeted. Although relevant clinical trials are relatively few and most do not concern IM, they can be used to demonstrate the important fact that neither CQ nor HCQ is used to enhance the antitumor efficacy of a drug by inhibiting its sequestration in lysosomes. Rather, they are used to increase the cytotoxic effects of chemotherapy through the inhibition of autophagy in the clinic as well [102,103,104,105,106]. Importantly, CQ and HCQ also potentiate the cytotoxic effects of temozolomide and gemcitabine, which definitely do not fit into the category of hydrophobic weakly basic drugs [102,103,104]. The only relevant work found on this topic was a randomised phase II trial of IM alone versus IM and HCQ targeting residual Bcr-Abl+ leukaemia stem cells responsible for disease persistence in CML patients [106]. The modest improvement in the combined therapy was attributed to autophagy inhibition via HCQ [106]. 

## 6. What Conclusions Can Be Drawn from Clinical Practice?

More than 20 years of experience using IM in clinical practice for the treatment of CML, GISTs, and other malignancies has revealed various mechanisms of resistance, none of which are related to lysosomes. The expression of ABCA3 was never found to be associated with a resistant phenotype in CML patients [96,97,98]. CQ and HCQ are used as inhibitors of autophagy and not as inhibitors of lysosomal sequestration to increase the antiproliferative effects of chemotherapy in clinical practice [102,103,104,105,106].

## 7. Conclusions

The analysis of the in vitro results shows that, even in the case of well-defined laboratory conditions, we have no clear evidence that the passive or enhanced lysosomal sequestration of IM leads to its reduced concentration in target sites and thus induces resistance to it. Similarly, the results of clinical trials do not indicate that lysosomal sequestration compromises IM therapeutic efficacy. If we consider the physicochemical properties of IM and its ability to massively accumulate in lysosomes, then it is appropriate to ask whether IM represents a proverbial exception that confirms the rule, or, on the contrary, it would be appropriate to critically re-evaluate this resistance mechanism.

## Figures and Tables

**Figure 1 cells-12-00709-f001:**
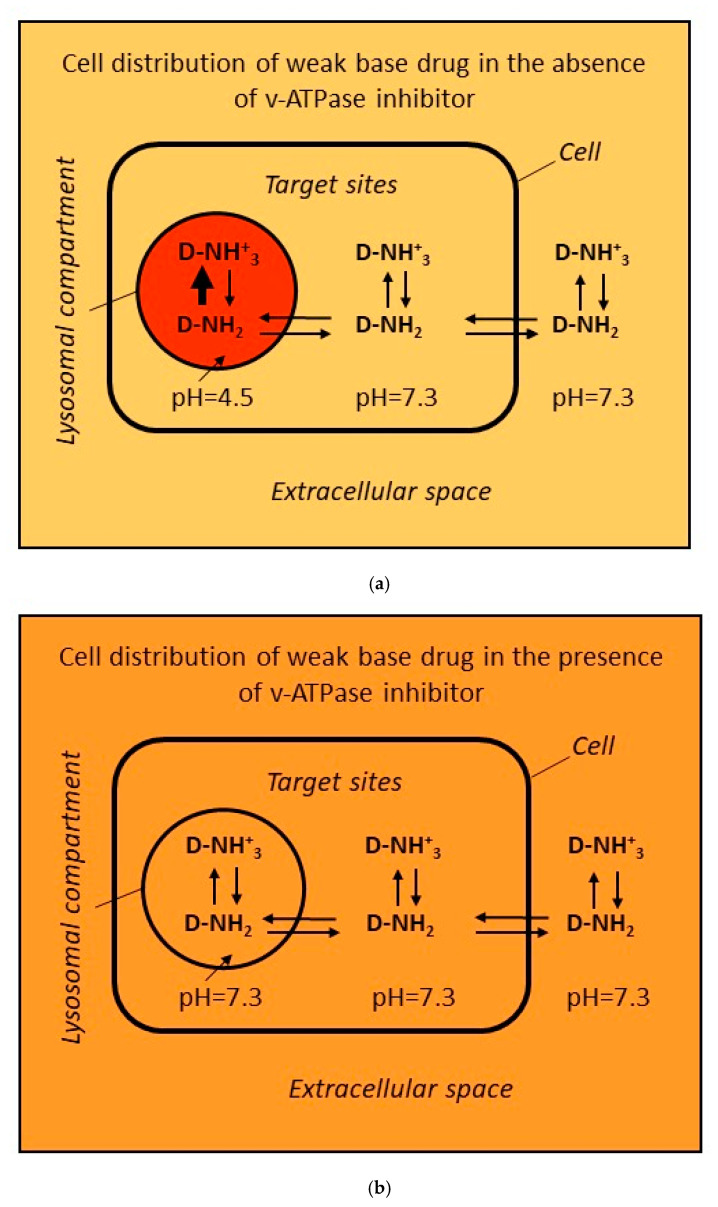
Passive lysosomal sequestration of weak-base drugs as a putative mechanism of drug resistance in vitro. Weak-base drug distribution is shown within the cell in vitro. This model envisages three simplifications: (i) only uncharged forms of a drug can freely diffuse across cellular membranes; (ii) only two interactions are considered, the Henderson–Hasselbach equilibrium ↑↓ and passive diffusion of uncharged molecules ⇆; (iii) for simplicity, we assumed the following pH setting: in lysosomes, the pH = 4.5 and in the extracellular (=growth medium) and extralysosomal (=cytosol) space, the same pH = 7.3. Colour saturation represents its concentration: the lighter the shade, the lower the concentration, and vice versa: (**a**) Weak-base drug distribution with passive lysosomal sequestration. Lysosomal sequestration is caused by the difference in pH between the lysosomes and the cytosol. A quantitative analysis of the IM distribution in the thought experiment is shown below. (**b**) Weak-base drug distribution without passive lysosomal sequestration. This can be achieved, for example, using BafA1, a vacuolar ATPase inhibitor. pH equalisation between lysosomes and cytosol leads to equal distribution of the hydrophobic weak-base drug in all compartments. (**c**) Graphic expression of drug resistance mediated by passive lysosomal sequestration. This mechanism of resistance assumes that there is a significant difference in the drug concentration at the target site between states when sequestration occurs and when it is prevented. This condition is then manifested by the different sensitivity to the drug; see the graph: red-dotted curve versus blue-dotted curve.

**Figure 2 cells-12-00709-f002:**
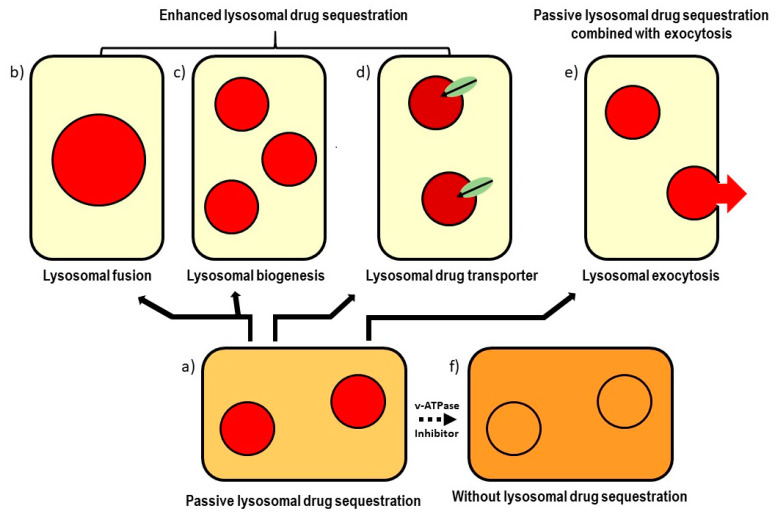
Putative mechanisms of lysosomal-mediated drug resistance. Passive lysosomal sequestration (**a**) can be enhanced by enlargement of the lysosomal compartment, through either lysosomal fusion (**b**) or lysosomal biogenesis (**c**). Passive lysosomal sequestration can be also enhanced by the functioning of lysosomal transporters (**d**). The combination of passive lysosomal sequestration with lysosomal exocytosis represents an additional putative mechanism of lysosomal-mediated drug resistance (**e**). Passive lysosomal sequestration can be reversed using v-ATPase inhibitors (**f**). Colour saturation represents its concentration: the lighter the shade, the lower the concentration, and vice versa.

**Table 1 cells-12-00709-t001:** Physicochemical properties of some conventional and targeted anticancer drugs.

Drug Name	Log *p*	pKa
Daunorubicin (DNR)	1.36	10.03
Doxorubicin (DOX)	0.54	10.03
Mitoxantrone (MTX)	0.65	9.36
Vinblastine (VNB)	4.18	8.86
Vincristine (VNC)	3.13	8.66
Imatinib (IM)	4.38	7.84
Dasatinib (DAS)	4.01	7.19
Bosutinb (BOS)	4.09	8.03
Ponatinib (PON)	4.97	7.62
Sunitinib (SUN)	2.93	9.04
Nintedanib (NTD)	2.79	7.23
Gefitinib (GEF)	3.75	6.85

Log *p* values were predicted using ChemAxon software and were adapted from DrugBank. pKa values referring to the strongest basic residue in each molecule were predicted using ChemAxon software and were adapted from DrugBank.

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
