# Peer review of "What Is the Significance of Lysosomal-Mediated Resistance to Imatinib?"

_cells, 2023, doi:10.3390/cells12050709_

Round 1

Reviewer 1 Report

Dear Editor,

The manuscript entitled “What is the significance of lysosomal mediated resistance to imatinib?” by Petr Mlejnek deals with the resistance mechanism showed by imatinib somehow in cells and humans, displaying a rational analysis of lysosome passive accumulation and possible correlation with resistance phenomena. The idea of rationally discussing this topic is original, but, on the whole, I am not sure that this can be treated as an isolated matter, strictly dependent on the in vitro intracellular trafficking of the drug. In addition, I guess that lysosomal sequestration of hydrophobic weak-base anti-cancer drugs is only due to passive accumulation in acid compartments where they are simply much more soluble! The peculiar molecular pathway observed in some works described can be a possible effect of a such accumulation instead of the cause. I suggest the author improving the discussion and conclude with a useful point of view in order to find a different viewpoint to inspire researchers.  

I suggest major revisions before publishing this manuscript:

1) The authors discussed the role of ABCA3 transporter in the capture of IM inside lysosome, but they also stated that <<Although these results are very convincing, the authors 109 did not provide direct evidence that ABCA3 transporter-enhanced lysosomal sequestra- 110 tion of IM reduces its concentration at the target sites. In my opinion there others aspects that should be discussed analytically that can explain well why IM is captures by lysosomes. For instance, the extremely high logP and low pKa (near 7.8) imply a very low solubility under physiological conditions and higher solubility under acidic conditions. Is this a reason which explain well the passive accumulation showed inside intracellular acid compartments?

2) Are the authors sure that clinical resistance to IM is not correlated to its poor in vivo biodistribution? Are there clinical studies about that? What is the typical pharmacokinetic of this drug after intramuscular or oral administration? As they are, without a clear correlation with biodistribution in humans, the authors’ comments are useless!

Author Response

Reviewer 1

Dear Editor,

The manuscript entitled “What is the significance of lysosomal mediated resistance to imatinib?” by Petr Mlejnek deals with the resistance mechanism showed by imatinib somehow in cells and humans, displaying a rational analysis of lysosome passive accumulation and possible correlation with resistance phenomena. The idea of rationally discussing this topic is original, but, on the whole, I am not sure that this can be treated as an isolated matter, strictly dependent on the in vitro intracellular trafficking of the drug. In addition, I guess that lysosomal sequestration of hydrophobic weak-base anti-cancer drugs is only due to passive accumulation in acid compartments where they are simply much more soluble! The peculiar molecular pathway observed in some works described can be a possible effect of a such accumulation instead of the cause. I suggest the author improving the discussion and conclude with a useful point of view in order to find a different viewpoint to inspire researchers.  

Answer:

I would like to thank Reviewer 1 very much for his/her stimulating comments on my review article. I substantially rewrote the review and added yours and other aspects in order to better clarify my point of view on the studied issue to the readers. The changes are in yellow.

I suggest major revisions before publishing this manuscript:

1) The authors discussed the role of ABCA3 transporter in the capture of IM inside lysosome, but they also stated that <<Although these results are very convincing, the authors 109 did not provide direct evidence that ABCA3 transporter-enhanced lysosomal sequestra- 110 tion of IM reduces its concentration at the target sites. In my opinion there others aspects that should be discussed analytically that can explain well why IM is captures by lysosomes. For instance, the extremely high logP and low pKa (near 7.8) imply a very low solubility under physiological conditions and higher solubility under acidic conditions. Is this a reason which explain well the passive accumulation showed inside intracellular acid compartments?

Answer:

Thanks to Reviewer 1 for the stimulating comment. I have substantially rewritten and expanded the MS to make it as understandable as possible to readers. Among other things, I have added images and a thought experiment that not only illustrate lysosomal sequestration, but also provide a quantitative analysis of this process. Please, see Section 3.1 and Figures 1 and 2 in the revised MS.

Solubility is relatively low 0.0146g/l (Drugbank) and acidic pH in lysosomes increases the solubility of IM and thus facilitate its extensive accumulation in this compartment -this is mentioned in section 3.1, point D in revised MS.  

2) Are the authors sure that clinical resistance to IM is not correlated to its poor in vivo biodistribution? Are there clinical studies about that? What is the typical pharmacokinetic of this drug after intramuscular or oral administration? As they are, without a clear correlation with biodistribution in humans, the authors’ comments are useless!

Answer:

I will try to answer sentence by sentence. To the best of my knowledge, there is no study that correlates clinical resistance to IM with its poor in vivo biodistribution. IM is administrated orally. Pharmacokinetic parameters of IM are very well summarized in review article by Peng et al (Peng P., Lloyd P., Schran H. Clinical Pharmacokinetics of Imatinib - Clin Pharmacokinet. 2005;44(9):879-894). I don't know exactly what you mean, however the PK parameters of IM do not show any peculiarities...

Resistance to chemotherapeutic agents is a widespread problem in cancer treatment. Whilst the outcomes are of prime importance in vivo, investigation of the mechanism usually involves in vitro experiments. Reversible binding and sequestration are important and need to be taken into account in the static in vitro situation. However, there is a fundamental difference between the dynamic in vivo and in vitro situation. Reversible binding and sequestration are irrelevant under steady state, serial dosing conditions (which applies to almost all oncology pharmaceutical interventions) and cannot be mechanisms of resistance. In this situation, the extracellular free drug concentration is controlled by intrinsic clearance, which in turn controls the intracellular drug concentration, influenced also by the membrane permeability of the drug and the presence of transporter systems (Smith, Di, Kerns. Nat. Rev. Drug Discov. 2010, 9, 929–939; Smith, Rowland. Drug Metab. Dispos. 2019, 47, 665–672).

Your last sentence touches me personally. I do not think that the analysis presented in my review, which is based on more than 100 published studies on the subject, both in vitro and clinical, is useless!

Reviewer 2 Report

Tyrosine kinase inhibitors (TKIs) are being increasingly used to treat various malignancies, they are also intimately linked with the mechanisms of multidrug resistance (MDR) in cancer cells.

 MDR-related solute carrier (SLC) and ATB-binding cassette (ABC) transporters are responsible for TKI uptake and efflux, respectively. However, the role of TKIs appears to be dual because they can act as substrates and/or inhibitors of these transporters. In addition, several TKIs have been identified to be sequestered into lysosomes either due to their physiochemical properties or via ABC transporters expressed on the lysosomal membrane. 

Furthermore i inform the authors that  the following topics should be further developed in the work:

1 Lysosomal Sequestration

Sequestration of TKIs into lysosomes provides a mechanism of resistance to TKIs

 2  The Mechanism  of TKIs on Membrane Transporters,  infact  the  TKIs do not harbor physiochemical properties of hydrophobic, weak base molecules but can be entrapped in the acidic milieu of lysosomes . ABC transporters facilitate the active accumulation of drugs in lysosomes, as these pumps have been found on the membranes of intracellular compartments, including the Golgi apparatus and intracellular vesicles.

 Many  authors  have  noted   that ABCA3 , ABCB1, and ABCG2 were demonstrated on lysosomal membranes, explaining the lysosomal sequestration of their respective substrate TKIs, including imatinib, sorafenib, and pazopanib 

 Furthermore, stressors present in the tumor microenvironment (e.g., hypoxia, oxidants, or glucose starvation) were found to upregulate and relocalize ABCB1 to lysosomal membranes, resulting in increased drug resistance

Author Response

Reviewer 2

Tyrosine kinase inhibitors (TKIs) are being increasingly used to treat various malignancies, they are also intimately linked with the mechanisms of multidrug resistance (MDR) in cancer cells.

 MDR-related solute carrier (SLC) and ATB-binding cassette (ABC) transporters are responsible for TKI uptake and efflux, respectively. However, the role of TKIs appears to be dual because they can act as substrates and/or inhibitors of these transporters. In addition, several TKIs have been identified to be sequestered into lysosomes either due to their physiochemical properties or via ABC transporters expressed on the lysosomal membrane. 

Answer:

I would like to thank Reviewer 2 very much for his/her stimulating comments on my review article. I substantially rewrote the review and added yours and other aspects in order to better clarify my point of view on the studied issue to the readers. The changes are in green.

Furthermore i inform the authors that  the following topics should be further developed in the work:

1 Lysosomal Sequestration

Sequestration of TKIs into lysosomes provides a mechanism of resistance to TKIs

Answer:

Thank you very much for the stimulating comment. I have added a paragraph 3.1. explaining the mechanism of lysosomal sequestration along with two figures (Fig 1 and 2). I have also added a thought experiment demonstrating quantitative parameters of lysosomal sequestration for IM in K562 chronic myeloid leukemia cells. Please, see revised MS.

 2  The Mechanism  of TKIs on Membrane Transporters,  infact  the  TKIs do not harbor physiochemical properties of hydrophobic, weak base molecules but can be entrapped in the acidic milieu of lysosomes . ABC transporters facilitate the active accumulation of drugs in lysosomes, as these pumps have been found on the membranes of intracellular compartments, including the Golgi apparatus and intracellular vesicles.

 Many  authors  have  noted   that ABCA3 , ABCB1, and ABCG2 were demonstrated on lysosomal membranes, explaining the lysosomal sequestration of their respective substrate TKIs, including imatinib, sorafenib, and pazopanib 

 Furthermore, stressors present in the tumor microenvironment (e.g., hypoxia, oxidants, or glucose starvation) were found to upregulate and relocalize ABCB1 to lysosomal membranes, resulting in increased drug resistence

Answer:

I dare to disagree with you on this point. Because the physicochemical characteristics of IM are as follows (logP=4.38; pKa=7.84, Tab. 1 in revised MS) I can't say that this is not a hydrophobic weak base. In addition, I deal in great detail with the issue of lysosomal ABC transporters. Please, see section 3.3 and section 5.3 in a revised version of my manuscript.

Reviewer 3 Report

This was a very interesting and well written short review covering the topic of whether lysosome sequestration of the weak base drug imatinib is related to the mechanism(s) causing drug resistance. Particularly interesting is the extent to which laboratory-based studies are relevant to clinical efficacy. 

In section 3.1 they wrote about studies that demonstrated specific mechanisms for drug resistance. However, those studies did not specifically examine lysosome sequestration-related mechanisms. Therefore, I think they need to “tone down” the last sentence of that section. 

In section 3.3 it is amazing that a drug can be sequestered by 70-80%, but yet have no effect on oncogenic signaling. It would be helpful if they could provide some additional insight into this paradox. 

Also, in section 3.3 it might be informative to review some literature in which the effects of lysosome de-acidification on lysosome sizes and numbers were studied. They could find such literature describing effects of other weak base drugs such as chloroquine and hydroxychloroquine as well as v-ATPase inhibitors.

Author Response

Reviewer 3

This was a very interesting and well written short review covering the topic of whether lysosome sequestration of the weak base drug imatinib is related to the mechanism(s) causing drug resistance. Particularly interesting is the extent to which laboratory-based studies are relevant to clinical efficacy. 

Answer:

I am very grateful to Reviewer 3 for his/her positive evaluation of my review and for very stimulating comments. I rewrote the review substantially mainly according to your comments and further add some new aspects in order to better clarify my point of view on the studied issue for the readers. The changes are in cyan.

In section 3.1 they wrote about studies that demonstrated specific mechanisms for drug resistance. However, those studies did not specifically examine lysosome sequestration-related mechanisms. Therefore, I think they need to “tone down” the last sentence of that section. 

Answer:

While I understand your comment, I'm going to take issue with you a bit on this point. On a general level, I would like to argue that if science only investigated phenomena that can be predicted in advance, we would probably still be somewhere at the beginning of knowledge... Regarding the practical level of your objection, I must note that at the time when mechanisms of resistance to IM were being sought, mechanisms related to lysosomal sequestration were also well known. So I can't agree with you. However, the corresponding part has been rewritten and expanded with additional references. In rewritten version the section 3.1 is now 3.2. Please, see the revised version of my MS. I have no doubt that all existing resistance mechanisms have been sought, including lysosome-mediated resistance. However, I have not found any study (either in vitro or clinical) that unequivocally identifies lysosomal sequestration as a mechanism of resistance to IM. Therefore, I do not consider my statement exaggerated.

In section 3.3 it is amazing that a drug can be sequestered by 70-80%, but yet have no effect on oncogenic signaling. It would be helpful if they could provide some additional insight into this paradox. 

Answer:

Thank you for the very interesting comment, although I think it should be addressed primarily to Burger et al Mol Pharmacol 2015. I don't think it's a real paradox. As I show in the added pictures (Figs. 1 and 2), in the thought experiment, and in the section 3.1 (please, see the revised version of my MS), the sequestration would have to be truly extreme for even the extracellular concentration to be significantly reduced (the uncharged form of the drug is in equilibrium in all compartments). And we don't know that because we only know the relative lysosomal accumulation (70-80%), but not the absolute lysosomal accumulation, the absolute cellular accumulation, and the total amount of IM in the system. Importantly, the measurements of Burger et al, ours, and other authors clearly show that inhibition of sequestration does not have a significant effect on oncogenic signaling i.e., lysosomal sequestration has marginal effect on drug concentration at target sites...

One more little note. You are right that the relative accumulation is very high (we measured lower). Every measurement is simply loaded with error...

Also, in section 3.3 it might be informative to review some literature in which the effects of lysosome de-acidification on lysosome sizes and numbers were studied. They could find such literature describing effects of other weak base drugs such as chloroquine and hydroxychloroquine as well as v-ATPase inhibitors.

Answer:

I would like to thank Reviewer 3 very much for this stimulating comment. According to your suggestions, I have extended my MS and added this aspect and discussed it in the context of possible mechanisms of drug resistance (paragraphs 3.5. and 5.4.). Please, see revised MS.

Round 2

Reviewer 1 Report

The authors have substantially answered to the Reviewer comment and, thus, the manuscript can be accepted as it is. 

Author Response

Thank you very much!

Petr Mlejnek

Reviewer 2 Report

I have read and re-read the work, I had asked the author to develop some concepts but not to write a new work

Paragraph 2

2.1.Lysosomal sequestration of imatinib and drug resistance in vitro

At the beginning of this review,

I will try to demonstrate with a thought experiment how much IM "this sentence must have been corrected" with the following

My approach was based on the following hypothesis

when you write Bcr-Abl+ K562 cells this had to be supported by in vitro experiments (Cell culture,Assay for the determination of intracellular IM levels,Western blot analysis)

I advise the author for this part to follow the following work:

The Lysosomal Sequestration of Tyrosine Kinase Inhibitors and Drug Resistance Eliska Ruzickova, Nikola Skoupa, Petr Dolezel, Dennis A. Smith, and Petr Mlejnek. Biomolecules. 2019 Nov; 9(11): 675

 Figures 1a and 1b need reference as they have already been published

2.2.Selection of Bcr-Abl positive cell lines with decreased sensitivity to imatinib

Although research on resistant cells selected with gradually increasing concentrations of IM has continued and yielded  some new findings elucidating changes associated with the development of resistance [64, 65], to the best of my knowledge no IM  resistant cells have been generated whose mechanism of resistance was lysosome-mediated "tthis sentence must be corrected as following"

some autors  have  demostrated that even though the studied TKIs, including imatinib, nilotinib, and dasatinib, were extensively accumulated in the lysosomes of cancer cells, their sequestration was insufficient to substantially reduce the extracellular drug concentration. Lysosomal accumulation of TKIs also failed to affect the Bcr-Abl signaling. . Importantly, even increased lysosomal sequestration of TKIs neither decreased their extracellular concentrations nor affected the sensitivity of Bcr-Abl to TKIs. In conclusion, the results of the work of autor Ruzickova clearly show that the lysosomal sequestration of TKIs failed to change their concentrations at target sites, and thus, can hardly contribute to drug resistance in vitro.

2.3.ABCA3 enhanced lysosomal sequestration of imatinib decreases sensitivity to this drug

ABCA3 has never been found as a possible cause of resistance to IM. Unfortunately, the answer to the question why it was  impossible to create CML cells with this resistance mechanism remains open.

In contrary other study have demostrated that cellular drug resistance partly relies on active transport across cellular membranes undertaken by ATP-binding cassette (ABC) transport proteins. Of those, the ABCA3 transporter has been found particularly overexpressed in AML cells

I suggest you read the following articles :

Prognostic impact of ABCA3 expression in adult and pediatric acute myeloid leukemia: an ALFA-ELAM02 joint studyAntony Ceraulo Blood Adv. 2022 May 10; 6(9): 2773–2777

ABC transporter A3 facilitates lysosomal sequestration of imatinib and modulates susceptibility of chronic myeloid leukemia cell lines to this drug, Bjoern Chapuy Vol. 94 No. 11 (2009): November, 2009 https://doi.org/10.3324/haematol.2009.008631

 However, to the best of my knowledge, no study has  yet reported ABCB1 expressed on the lysosomal membrane as a cause of IM resistance .

 Furthermore, as have  writed  Maria Krchniakova and  collegues   in the   work  that   ABCB1 [88], and ABCG2 [89] were demonstrated on lysosomal membranes, explaining the lysosomal sequestration of their respective substrate TKIs, including imatinib , sorafenib  and pazopanib.Interestingly, the ABCB1-mediated resistance phenotype of leukemia cells was stronger when ABCB1 was expressed intracellularly than when it was expressed on the plasma membrane, indicating that the accumulation of drugs in lysosomes is most likely more effective than the efflux via membrane transporters. Furthermore, stressors present in the tumor microenvironment (e.g., hypoxia, oxidants, or glucose starvation) were found to upregulate and relocalize ABCB1 to lysosomal membranes, resulting in increased drug resistance

I suggest you read the following article :

Repurposing Tyrosine Kinase Inhibitors to Overcome Multidrug Resistance in Cancer: Focus on Transporters and Lysosomal Sequestration by Maria Krchniakova Int. J. Mol. Sci. 2020, 21(9), 3157

 I have read and re-read the work, I had asked the author to develop some concepts but not to write a new work

Paragraph 2

2.1.Lysosomal sequestration of imatinib and drug resistance in vitro

At the beginning of this review,

I will try to demonstrate with a thought experiment how much IM "this sentence must have been corrected" with the following

My approach was based on the following hypothesis

when you write Bcr-Abl+ K562 cells this had to be supported by in vitro experiments (Cell culture,Assay for the determination of intracellular IM levels,Western blot analysis)

I advise the author for this part to follow the following work:

The Lysosomal Sequestration of Tyrosine Kinase Inhibitors and Drug Resistance Eliska Ruzickova, Nikola Skoupa, Petr Dolezel, Dennis A. Smith, and Petr Mlejnek. Biomolecules. 2019 Nov; 9(11): 675

 Figures 1a and 1b need reference as they have already been published

2.2.Selection of Bcr-Abl positive cell lines with decreased sensitivity to imatinib

Although research on resistant cells selected with gradually increasing concentrations of IM has continued and yielded  some new findings elucidating changes associated with the development of resistance [64, 65], to the best of my knowledge no IM  resistant cells have been generated whose mechanism of resistance was lysosome-mediated "tthis sentence must be corrected as following"

some autors  have  demostrated that even though the studied TKIs, including imatinib, nilotinib, and dasatinib, were extensively accumulated in the lysosomes of cancer cells, their sequestration was insufficient to substantially reduce the extracellular drug concentration. Lysosomal accumulation of TKIs also failed to affect the Bcr-Abl signaling. . Importantly, even increased lysosomal sequestration of TKIs neither decreased their extracellular concentrations nor affected the sensitivity of Bcr-Abl to TKIs. In conclusion, the results of the work of autor Ruzickova clearly show that the lysosomal sequestration of TKIs failed to change their concentrations at target sites, and thus, can hardly contribute to drug resistance in vitro.

2.3.ABCA3 enhanced lysosomal sequestration of imatinib decreases sensitivity to this drug

ABCA3 has never been found as a possible cause of resistance to IM. Unfortunately, the answer to the question why it was  impossible to create CML cells with this resistance mechanism remains open.

In contrary other study have demostrated that cellular drug resistance partly relies on active transport across cellular membranes undertaken by ATP-binding cassette (ABC) transport proteins. Of those, the ABCA3 transporter has been found particularly overexpressed in AML cells

I suggest you read the following articles :

Prognostic impact of ABCA3 expression in adult and pediatric acute myeloid leukemia: an ALFA-ELAM02 joint studyAntony Ceraulo Blood Adv. 2022 May 10; 6(9): 2773–2777

ABC transporter A3 facilitates lysosomal sequestration of imatinib and modulates susceptibility of chronic myeloid leukemia cell lines to this drug, Bjoern Chapuy Vol. 94 No. 11 (2009): November, 2009 https://doi.org/10.3324/haematol.2009.008631

 However, to the best of my knowledge, no study has  yet reported ABCB1 expressed on the lysosomal membrane as a cause of IM resistance .

 Furthermore, as have  writed  Maria Krchniakova and  collegues   in the   work  that   ABCB1 [88], and ABCG2 [89] were demonstrated on lysosomal membranes, explaining the lysosomal sequestration of their respective substrate TKIs, including imatinib , sorafenib  and pazopanib.Interestingly, the ABCB1-mediated resistance phenotype of leukemia cells was stronger when ABCB1 was expressed intracellularly than when it was expressed on the plasma membrane, indicating that the accumulation of drugs in lysosomes is most likely more effective than the efflux via membrane transporters. Furthermore, stressors present in the tumor microenvironment (e.g., hypoxia, oxidants, or glucose starvation) were found to upregulate and relocalize ABCB1 to lysosomal membranes, resulting in increased drug resistance

I suggest you read the following article :

Repurposing Tyrosine Kinase Inhibitors to Overcome Multidrug Resistance in Cancer: Focus on Transporters and Lysosomal Sequestration by Maria Krchniakova Int. J. Mol. Sci. 2020, 21(9), 3157

 I have read and re-read the work, I had asked the author to develop some concepts but not to write a new work

Paragraph 2

2.1.Lysosomal sequestration of imatinib and drug resistance in vitro

At the beginning of this review,

I will try to demonstrate with a thought experiment how much IM "this sentence must have been corrected" with the following

My approach was based on the following hypothesis

when you write Bcr-Abl+ K562 cells this had to be supported by in vitro experiments (Cell culture,Assay for the determination of intracellular IM levels,Western blot analysis)

I advise the author for this part to follow the following work:

The Lysosomal Sequestration of Tyrosine Kinase Inhibitors and Drug Resistance Eliska Ruzickova, Nikola Skoupa, Petr Dolezel, Dennis A. Smith, and Petr Mlejnek. Biomolecules. 2019 Nov; 9(11): 675

 Figures 1a and 1b need reference as they have already been published

2.2.Selection of Bcr-Abl positive cell lines with decreased sensitivity to imatinib 

Although research on resistant cells selected with gradually increasing concentrations of IM has continued and yielded  some new findings elucidating changes associated with the development of resistance [64, 65], to the best of my knowledge no IM  resistant cells have been generated whose mechanism of resistance was lysosome-mediated "tthis sentence must be corrected as following"

some autors  have  demostrated that even though the studied TKIs, including imatinib, nilotinib, and dasatinib, were extensively accumulated in the lysosomes of cancer cells, their sequestration was insufficient to substantially reduce the extracellular drug concentration. Lysosomal accumulation of TKIs also failed to affect the Bcr-Abl signaling. . Importantly, even increased lysosomal sequestration of TKIs neither decreased their extracellular concentrations nor affected the sensitivity of Bcr-Abl to TKIs. In conclusion, the results of the work of autor Ruzickova clearly show that the lysosomal sequestration of TKIs failed to change their concentrations at target sites, and thus, can hardly contribute to drug resistance in vitro. 

2.3.ABCA3 enhanced lysosomal sequestration of imatinib decreases sensitivity to this drug

ABCA3 has never been found as a possible cause of resistance to IM. Unfortunately, the answer to the question why it was  impossible to create CML cells with this resistance mechanism remains open.

In contrary other study have demostrated that cellular drug resistance partly relies on active transport across cellular membranes undertaken by ATP-binding cassette (ABC) transport proteins. Of those, the ABCA3 transporter has been found particularly overexpressed in AML cells

I suggest you read the following articles :

Prognostic impact of ABCA3 expression in adult and pediatric acute myeloid leukemia: an ALFA-ELAM02 joint studyAntony Ceraulo Blood Adv. 2022 May 10; 6(9): 2773–2777

ABC transporter A3 facilitates lysosomal sequestration of imatinib and modulates susceptibility of chronic myeloid leukemia cell lines to this drug, Bjoern Chapuy Vol. 94 No. 11 (2009): November, 2009 https://doi.org/10.3324/haematol.2009.008631

 However, to the best of my knowledge, no study has  yet reported ABCB1 expressed on the lysosomal membrane as a cause of IM resistance .

 Furthermore, as have  writed  Maria Krchniakova and  collegues   in the   work  that   ABCB1 [88], and ABCG2 [89] were demonstrated on lysosomal membranes, explaining the lysosomal sequestration of their respective substrate TKIs, including imatinib , sorafenib  and pazopanib.Interestingly, the ABCB1-mediated resistance phenotype of leukemia cells was stronger when ABCB1 was expressed intracellularly than when it was expressed on the plasma membrane, indicating that the accumulation of drugs in lysosomes is most likely more effective than the efflux via membrane transporters. Furthermore, stressors present in the tumor microenvironment (e.g., hypoxia, oxidants, or glucose starvation) were found to upregulate and relocalize ABCB1 to lysosomal membranes, resulting in increased drug resistance

I suggest you read the following article :

Repurposing Tyrosine Kinase Inhibitors to Overcome Multidrug Resistance in Cancer: Focus on Transporters and Lysosomal Sequestration by Maria Krchniakova Int. J. Mol. Sci. 2020, 21(9), 3157

Author Response

I have read and re-read the work, I had asked the author to develop some concepts but not to write a new work

Answer:

I would like to thank Reviewer 2 for his/her additional comments on my review article.

I cannot agree with your claim that I have written a new work. I edited the MS according to your suggestions, although it was difficult to understand what you actually wanted, since your comments consist largely of copied passages from Krchniakova et al. 2020.

I expanded the original work based on the comments of all three Reviewers in order to better clarify my point of view on the studied issue to the readers. The concept of work remained the same and I think it is clear.

 Paragraph 2

 2.1.Lysosomal sequestration of imatinib and drug resistance in vitro

 At the beginning of this review,

I will try to demonstrate with a thought experiment how much IM "this sentence must have been corrected" with the following

 My approach was based on the following hypothesis

Answer:

I do not accept your suggestion.

 when you write Bcr-Abl+ K562 cells this had to be supported by in vitro experiments (Cell culture,Assay for the determination of intracellular IM levels,Western blot analysis)

I advise the author for this part to follow the following work:

 The Lysosomal Sequestration of Tyrosine Kinase Inhibitors and Drug Resistance Eliska Ruzickova, Nikola Skoupa, Petr Dolezel, Dennis A. Smith, and Petr Mlejnek. Biomolecules. 2019 Nov; 9(11): 675

Answer:

I do not accept your suggestion as it is irrelevant.

 Figures 1a and 1b need reference as they have already been published

Answer:

I do not accept your suggestion. All these images are my own work and are very basic. Therefore, it makes no sense to refer to another study of ours. In this review, I present them to explain the principle of lysosomal sequestration and for a better understanding of the calculation (thought experiment).

 2.2.Selection of Bcr-Abl positive cell lines with decreased sensitivity to imatinib

Although research on resistant cells selected with gradually increasing concentrations of IM has continued and yielded  some new findings elucidating changes associated with the development of resistance [64, 65], to the best of my knowledge no IM  resistant cells have been generated whose mechanism of resistance was lysosome-mediated "tthis sentence must be corrected as following"

some autors  have  demostrated that even though the studied TKIs, including imatinib, nilotinib, and dasatinib, were extensively accumulated in the lysosomes of cancer cells, their sequestration was insufficient to substantially reduce the extracellular drug concentration. Lysosomal accumulation of TKIs also failed to affect the Bcr-Abl signaling. . Importantly, even increased lysosomal sequestration of TKIs neither decreased their extracellular concentrations nor affected the sensitivity of Bcr-Abl to TKIs. In conclusion, the results of the work of autor Ruzickova clearly show that the lysosomal sequestration of TKIs failed to change their concentrations at target sites, and thus, can hardly contribute to drug resistance in vitro.

Answer:

I cannot agree with your suggestion as it does not make sense. In addition, formulation you suggest me to write was copied from abstract of our study by Ruzickova et al., 2019 - ref [43]

2.3.ABCA3 enhanced lysosomal sequestration of imatinib decreases sensitivity to this drug

 ABCA3 has never been found as a possible cause of resistance to IM. Unfortunately, the answer to the question why it was  impossible to create CML cells with this resistance mechanism remains open.

 In contrary other study have demostrated that cellular drug resistance partly relies on active transport across cellular membranes undertaken by ATP-binding cassette (ABC) transport proteins. Of those, the ABCA3 transporter has been found particularly overexpressed in AML cells

 I suggest you read the following articles :

 Prognostic impact of ABCA3 expression in adult and pediatric acute myeloid leukemia: an ALFA-ELAM02 joint studyAntony Ceraulo Blood Adv. 2022 May 10; 6(9): 2773–2777

 ABC transporter A3 facilitates lysosomal sequestration of imatinib and modulates susceptibility of chronic myeloid leukemia cell lines to this drug, Bjoern Chapuy Vol. 94 No. 11 (2009): November, 2009 https://doi.org/10.3324/haematol.2009.008631

Answer:

I do not understand why you suggest me to read these articles…

First, in my review, the publication Chapuy et al 2009 - ref [33] is not only cited, but also widely discussed. Second, I cite and discuss here the work of Chapuy et al. 2008 and other related works from the laboratory of Professor G. Wulf, which also includes the Bjoern Chapuy you mentioned -ref [32, 66, 99]. Furthermore, I cite and discuss other relevant studies - ref [97, 98]. As for Antony Ceraulo's study, I mention three earlier studies which nevertheless came to exactly the same conclusions – ref [100-102].

Please, read my review again chapter 3.3. (the first two paragraphs) and chapter 5.3.

However, to the best of my knowledge, no study has  yet reported ABCB1 expressed on the lysosomal membrane as a cause of IM resistance .

Furthermore, as have  writed  Maria Krchniakova and  collegues   in the   work  that   ABCB1 [88], and ABCG2 [89] were demonstrated on lysosomal membranes, explaining the lysosomal sequestration of their respective substrate TKIs, including imatinib , sorafenib  and pazopanib.Interestingly, the ABCB1-mediated resistance phenotype of leukemia cells was stronger when ABCB1 was expressed intracellularly than when it was expressed on the plasma membrane, indicating that the accumulation of drugs in lysosomes is most likely more effective than the efflux via membrane transporters. Furthermore, stressors present in the tumor microenvironment (e.g., hypoxia, oxidants, or glucose starvation) were found to upregulate and relocalize ABCB1 to lysosomal membranes, resulting in increased drug resistance

 I suggest you read the following article :

Repurposing Tyrosine Kinase Inhibitors to Overcome Multidrug Resistance in Cancer: Focus on Transporters and Lysosomal Sequestration by Maria Krchniakova Int. J. Mol. Sci. 2020, 21(9), 3157

 Answer:

I stand by my conclusion and within the text in the third paragraph of chapter 3.3. I explain the reasons mainly based on the review article by Szakacz and Abele -ref [71], which states, among other things, that no one has yet proven that ABCB1 is functionally expressed on the lysosomal membrane.

I know Marie Kchniakova's article, including the passages you copy from this work. Unfortunately, this work does not bring any new ideas. In addition, there are inaccuracies, for example, the citation [89] does not refer to ABCG2 at all, but to ABCB1. In addition, the study cited herein as reference [84] did not find or conclude that expression of any of the ABC transporters on lysosomes mediates resistance to pazopanib. And many other inaccuracies can be found throughout the text. So, I see no reason to pursue this work.